# Properties of Micro-Arc Oxidation Coatings on 5052 Al Alloy Sealed by SiO₂ Nanoparticles

**Siqi Liu, Jiahuan Chen, Dongdong Zhang, Yuxin Wang \*, Zhen He \* and Pingyi Guo**

School of Materials Science and Engineering, Jiangsu University of Science and Technology, Zhenjiang 212003, China; lsq15376988099@163.com (S.L.); 15606107161@163.com (J.C.); zhangdong0820@just.edu.cn (D.Z.); pyguo@just.edu.cn (P.G.)

\* Correspondence: ywan943@163.com (Y.W.); hezhen@just.edu.cn (Z.H.)

**Abstract:** Micro-arc oxidation (MAO) treatment can effectively improve the wear resistance, corrosion resistance, and mechanical strength of aluminum alloy substrates. Improving the porous structure of MAO film and effectively sealing the pores is a significant research issue. In this study, the MAO treatment of 5052 aluminum alloy was carried out in silicate electrolytes. The MAO films were sealed with different concentrations of SiO₂ nanoparticles. The effects of SiO₂ nanoparticle content on the MAO films' microstructure, mechanical properties, and corrosion performance were systematically investigated. When adding SiO₂ nanoparticles to electrolytes, the particles were deposited at the micropores of the film, which could effectively seal the porous MAO film and significantly improve its corrosion and wear resistance. The corrosion resistance and wear resistance properties were optimal with 5.0 g/L SiO₂ addition. Compared to the unsealed film, the corrosion current density and corrosion rate decreased from $1.24 \times 10^{-9}$ A/cm² and $1.47 \times 10^{-5}$ mm/a to $7.78 \times 10^{-10}$ A/cm² and $9.15 \times 10^{-6}$ mm/a, respectively. Moreover, the average friction coefficient of the sealed film was 0.606, which was ~19.3% lower than that of the substrate and 3.3% lower than for the unsealed film.

**Keywords:** micro-arc oxidation (MAO); SiO₂ nanoparticles; corrosion resistance; wear property





## 1. Introduction

Aluminum and its alloy materials are widely used in automotive, aerospace, and other industrial applications because of their high specific strength, strong corrosion resistance, superior machinability, good thermal conductivity, and recyclability [1,2]. However, aluminum has some disadvantages as a structural material, such as soft texture, poor wear resistance, low corrosion resistance, and poor heat resistance, which seriously affect the scaled-up applications [3,4].

To improve the performance of aluminum alloy, various surface engineering methods have been developed and applied. Common surface treatment methods include anodic oxidation [5–7], electroplating [8,9], plasma spraying, laser surface modification, and micro-arc oxidation [10,11]. Micro-arc oxidation (MAO) is a novel surface treatment technology that allows in situ deposition of autogenous ceramic films on magnesium, aluminum, and titanium [12–14]. MAO films are characterized by solid adhesion, compact structure, high strength, exceptional corrosion resistance [15,16], and excellent high-temperature impact resistance. However, many microcracks and discharge micropores inevitably appear on MAO films due to the special reaction process [17–19]. The corrosive medium can easily penetrate the membrane and interface layers, leading to the corrosion of substrate material [20].

In addition to optimizing the MAO process, it is necessary to seal the pores of the MAO film and enhance the shielding effect in the corrosive medium [21,22]. It has been found that the corrosion resistance of MAO films can be improved by applying different sealing materials and sealing processes [23,24]. The traditional sealing methods can be divided

into hydrated, inorganic, and organic seals [25,26]. Inorganic seals have promising application prospects due to their simple process, low cost, and easy implementation [27–29]. Introducing highly stable oxide nanoparticles, such as $TiO_2$, $ZrO_2$, and $SiO_2$, can effectively enhance corrosion protection [30,31]. The suspension of $SiO_2$ nanoparticles can be used to prepare coatings with enhanced thermomechanical properties, reduced surface porosity, and improved wear resistance [32–34]. $SiO_2$ nanoparticles can act as sintering aids to promote the formation of MAO films [35–38]. Meanwhile, $SiO_2$ nanoparticles with high melting point and small size can reduce surface cracks and increase the density of ceramic layers, thus improving the corrosion and wear resistance of surface layers [39,40].

In the present study, the surface of 5052 Al alloy was treated by MAO process with different addition (0–10 g/L) of $SiO_2$ nanoparticles (particle size of 20 nm) in silicate electrolytes. The nanoparticles were uniformly suspended in the electrolyte by high-speed stirring and ultrasonic shaking to seal the generated pores. The effects of $SiO_2$ nanoparticle content on the microstructure, mechanical properties and corrosion resistance of MAO films were systematically investigated.

## 2. Experimental Method

### 2.1. Sample Preparation

The substrates used in this research were 5052 Al alloys with dimensions of 20 mm × 20 mm × 6 mm. The substrates were carefully pretreated before being subject to the MAO process. The samples were polished and ultrasonically cleaned in ethanol and deionized water for 5 min to remove the surface oil. The chemical composition of the 5052 Al alloy is detailed in Table 1.

**Table 1.** Chemical Composition of 5052 Al Alloy.

| Element | Si | Fe | Cu | Mn | Mg | Cr | Zn | Al |
|---|---|---|---|---|---|---|---|---|
| Content (wt.%) | 25 | 0.4 | 0.1 | 0.3 | 2.5 | 0.2 | 0.1 | Bal |

The MAO treatment was conducted using micro-arc oxidation equipment (WHD-30, Harbin, China). The electrolyte composition and process parameters are shown in Table 2. The electrolyte was prepared with deionized water. All reagents were purchased from the Sigma company with analytical purity. We used the bidirectional pulse current. The positive and negative current density was 6 A/dm$^2$. The positive and negative duty cycles were 30% & 40%.

**Table 2.** Electrolyte Composition and MAO Process Parameters.

| Reagents and Conditions | Values |
|---|---|
| $Na_2SiO_3 \cdot 9H_2O$ | 10 g/L |
| KOH | 1 g/L |
| $Na_2WO_4 \cdot 2H_2O$ | 1 g/L |
| $SiO_2$ nanoparticle | 0–10 g/L |
| pH | 12~14 |
| Current density | 6 A/dm$^2$ |
| Frequency | 400 Hz |
| Duty cycle | 30%/40% |
| Pulse ratio | 1/1 |
| Oxidation time | 10 mins |

### 2.2. Sample Characterization

The microstructure and chemical distribution of the MAO films was characterized by a scanning electron microscope (SEM, Phenom Pro X, Eindhoven, The Netherlands) equipped with an energy dispersive spectrometer (EDS). The phase structure of the MAO films was studied using an X-ray diffractometer (XRD, Shimadzu XRD-6000, Kyoto, Japan).

The electrochemical impedance spectroscopy and potentiodynamic polarization curve of the MAO films were tested using an electrochemical workstation (PGSTAT302 N, Metrohm Autolab B.V., Utrecht, The Netherlands). For the three-electrode corrosion cell, the counter electrode was made of platinum mesh, the reference electrode was a saturated calomel electrode, and the sample was the working electrode. The corrosion results were collected in 3.5 wt.% NaCl solution at room temperature. The microhardness of the MAO films was tested by a semi-automatic microhardness tester (HXS-1000 TAC, Shanghai, China). The load was 10 N and the loading time was set as 15 s. A reciprocating wear tester (HSR-2 M, Lanzhou, China) examined the friction properties of the MAO films. A $Si_3 N_4$ ceramic ball with a diameter of 4 mm was used for the test and the total sliding distance was set as 20 m for each sample. All the wear tests were conducted at room temperature for 10 mins under 5 N load.

### 3. Results and Discussion

#### *3.1. Microstructures*

The microstructures of the MAO films sealed using different concentrations of $SiO_2$ nanoparticles were observed, as presented in Figure 1. Figure 1a shows the microstructure of unsealed MAO film prepared by the common electrolyte. It can be clearly seen that there are dense "honeycomb" holes on the film surface. These holes provide channels for the flow of corrosive media. During the MAO process, the surface film layer is constantly broken down by high voltage. The molten $Al_2 O_3$ was ejected and cooled rapidly in the external electrolyte, resulting in the formation of holes. When 2.5 g/L $SiO_2$ nanoparticles were added into the electrolyte, $SiO_2$ nanoparticles were deposited outside the discharge micropores and a small amount of aggregation occurred, as shown in Figure 1b. When the addition of $SiO_2$ increased to 5 g/L, the micropores were more obviously covered by $SiO_2$, as shown in Figure 1c, thereby blocking the corrosion channel. However, when excess $SiO_2$ was added, as shown in Figure 1d–e, the nanoparticles gathered on the film surface under discharge, adsorbing a large number of free ions, and accentuating the micro-arc discharge, resulting in high surface energy and ablation, so that the molten oxide could not solidify well on the surface of the film. When the $SiO_2$ concentration reached 10.0 g/L, film roughness was higher and surface oxide ablation was accentuated.

The elemental content of MAO films sealed with different concentrations of $SiO_2$ nanoparticles was determined by EDS analysis, as shown in Table 3. The main elements in the MAO films prepared by common electrolyte were O, Al and Si. O was mainly derived from $SiO_3^{2-}$ in the electrolyte and occurred as $SiO_2$ and $Al_2 O_3$, while Al was derived from the matrix, and Si was derived from the residual $SiO_3^{2-}$ and $SiO_2$ generated by the discharge of $SiO_3^{2-}$. The atomic percentage of Si in the film increased continuously with addition of $SiO_2$ nanoparticles and reached 27.22% when the concentration of $SiO_2$ nanoparticles in the electrolyte was 10 g/L, confirming that the $SiO_2$ nanoparticles had entered the film layer.

**Table 3.** EDS Analysis (at%) of Film.

| Samples | O | Al | Si | W |
|---|---|---|---|---|
| Unsealed MAO film | 61.02 | 22.65 | 15.60 | 0.72 |
| MAO film sealed with 2.5 g/L $SiO_2$ | 63.80 | 12.13 | 20.97 | 3.10 |
| MAO film sealed with 5.0 g/L $SiO_2$ | 63.91 | 9.51 | 23.89 | 2.69 |
| MAO film sealed with 7.5 g/L $SiO_2$ | 65.86 | 6.25 | 25.55 | 2.34 |
| MAO film sealed with 10.0 g/L $SiO_2$ | 66.69 | 2.68 | 27.22 | 3.41 |

Figure 2 shows the cross-sectional morphology of MAO films prepared with different concentrations of $SiO_2$ nanoparticles. The results showed that the thickness of the film layer increased with increase in $SiO_2$ concentration. Figure 2a shows the MAO film layer without $SiO_2$ nanoparticle addition, having a thickness of ~16 μm. The dense area near the substrate had fewer pores, while the outer area had some pores penetrating the film. Figure 2b shows

the MAO film layer with 2.5 g/L $SiO_2$ addition, having a thickness of ~20 µm. Many cracks and large inner pores appeared in the film. Figure 2c shows the MAO film layer prepared by adding 5.0 g/L $SiO_2$ nanoparticles, having a thickness of ~33 µm. The internal dense area showed a certain degree of porosity with small pores, while cracks and holes were still evident. There were some particles deposited on the surface, blocking the micropores generated in the MAO process. Figure 2d shows the MAO film layer prepared by adding 7.5 g/L $SiO_2$. The thickness of the film continued to increase to ~45 µm, yet the number of inner pores increased with larger pore diameter. Figure 2e shows that there was no significant increase in thickness when the nanoparticle concentration was further increased. The larger pore size holes inside them can seriously damage corrosion resistance.

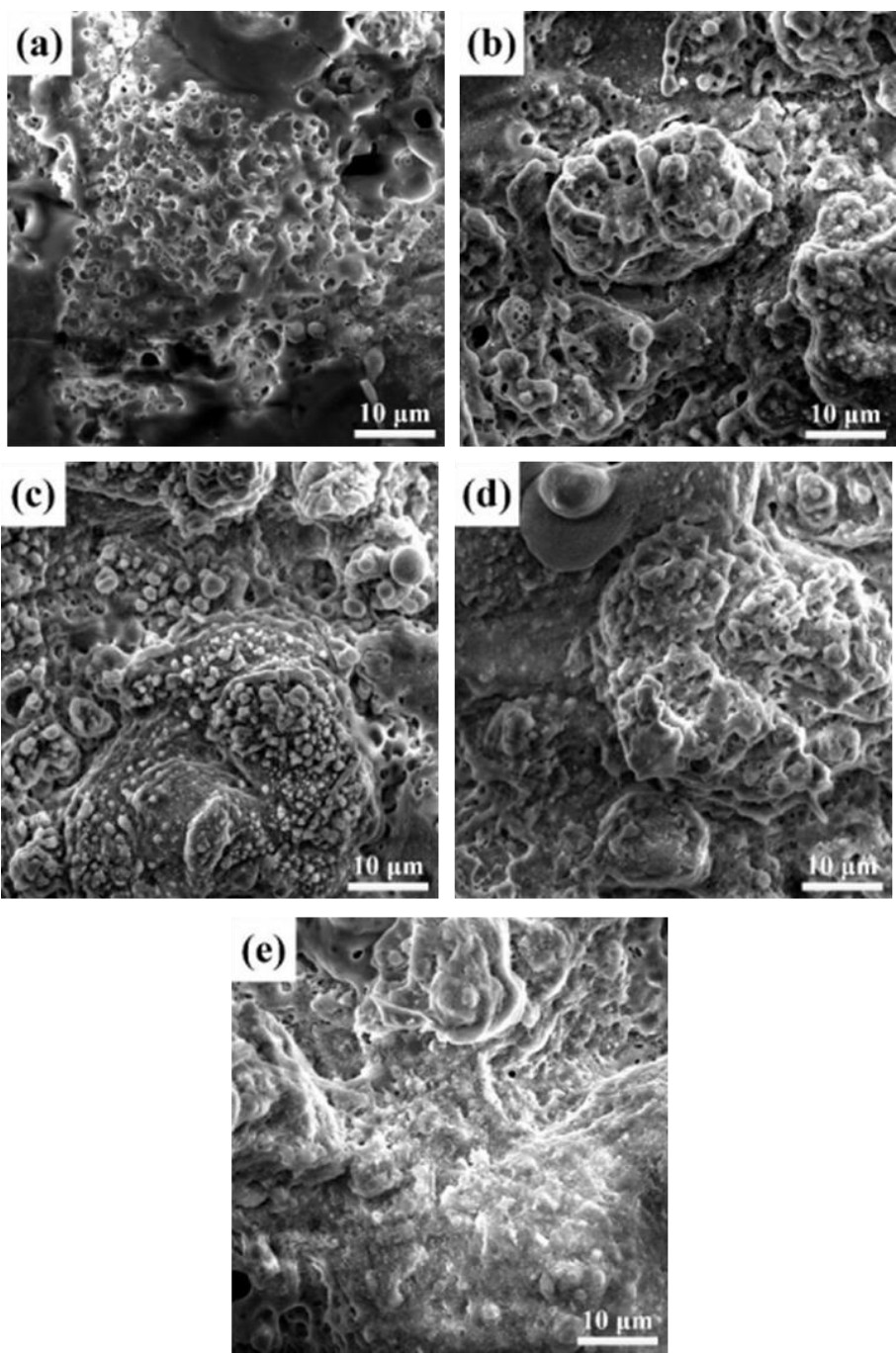

**Figure 1.** Surface SEM images of (**a**) unsealed MAO film and (**b**) MAO films sealed by different concentrations of $SiO_2$ nanoparticles: 2.5 g/L, (**c**) 5 g/L (**d**) 7.5 g/L and (**e**) 10 g/L.

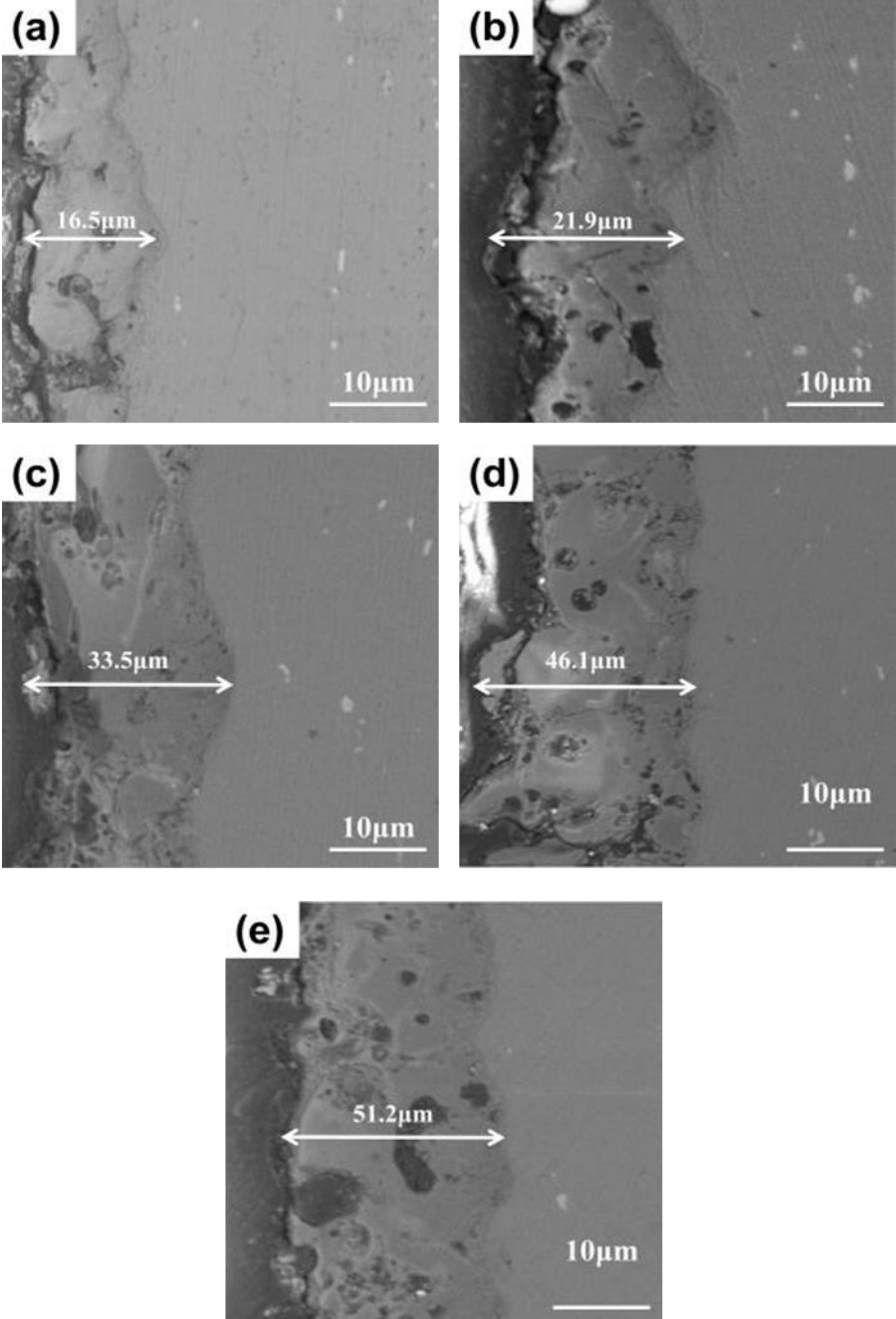

**Figure 2.** Cross-sectional SEM images of (**a**) unsealed MAO film and, (**b**) MAO films sealed by different concentrations of SiO$_2$ nanoparticles: 2.5 g/L, (**c**) 5 g/L (**d**) 7.5 g/L and (**e**) 10 g/L.

Figure 3 presents the cross-sectional elemental distribution of unsealed and 5.0 g/L SiO$_2$ sealed films. The MAO film prepared with the addition of 5 g/L nano-SiO$_2$ had a higher content and a more uniform distribution of Si compared to the unsealed layer. The Si concentrated in the outer sparse area, indicating that the nano-SiO$_2$ particles were well-deposited on the surface. The distribution density of Si increased in the pores and cracks. It can be inferred that SiO$_2$ nanoparticles were continuously deposited in the micropores, blocking the micropores and filling the cracks. Moreover, a certain amount of Si was also detected in the inner areas, implying that the SiO$_2$ nanoparticles were successfully incorporated into the MAO film.

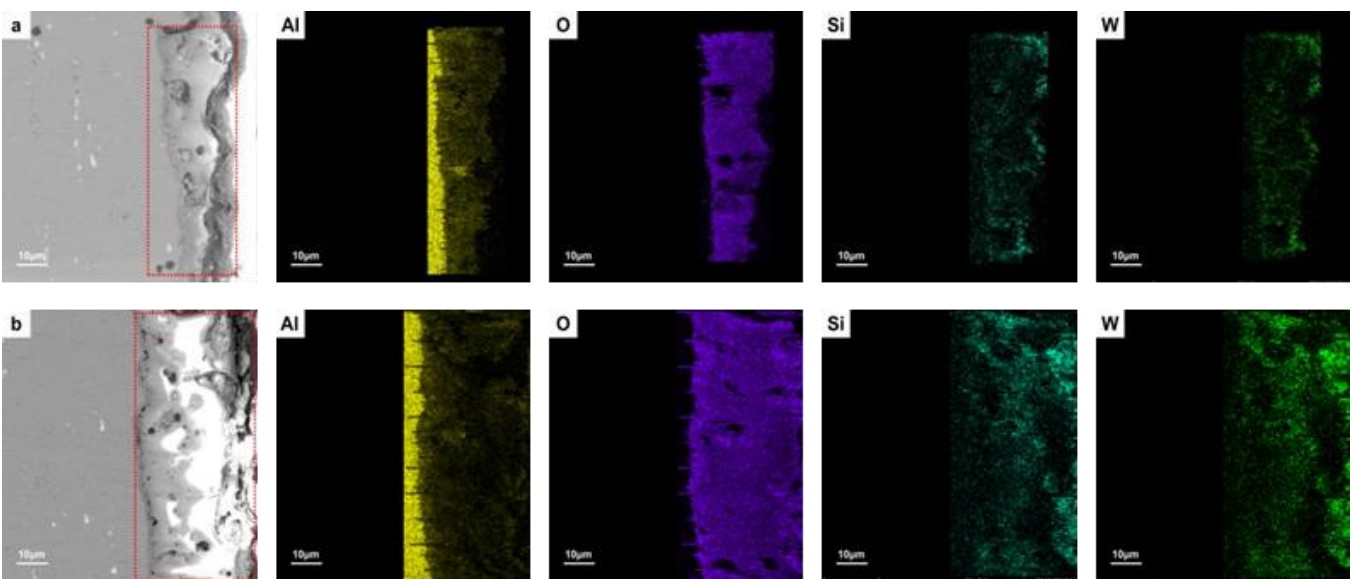

**Figure 3.** Cross-sectional images and element distribution of (**a**) unsealed MAO film, and (**b**) MAO film sealed with 5 g/L concentration of $SiO_2$ nanoparticles.

*3.2. XRD Analysis*

Figure 4 shows the XRD patterns of MAO thin films and MAO thin films sealed by $SiO_2$ nanoparticles. It can be seen from the figure that the phase structure of MAO thin films was mainly composed of Al, $\alpha$-$Al_2O_3$, $\gamma$-$Al_2O_3$ and mullite. The intensity of the diffraction peaks of Al was mainly from the matrix. During the MAO process, $Al_2O_3$ with different crystal structures was formed in the MAO films as a result of the combined effect of chemical and electrochemical reactions. When aluminum is placed in alkaline electrolyte, a dense oxide film is rapidly formed on the substrate surface. The formation of different crystalline types of $Al_2O_3$ was related to the cooling rate and cooling temperature. When the cooling rate was relatively fast, the molten $Al_2O_3$ was converted mostly to $\gamma$-$Al_2O_3$. When the cooling rate was relatively slow, the molten $Al_2O_3$ was converted mostly to $\alpha$-$Al_2O_3$, and some $\gamma$-$Al_2O_3$ was also converted to $\alpha$-$Al_2O_3$. At the same time, the presence of mullite peaks indicated the successful incorporation of nano-$SiO_2$ into the film layer during the MAO process.

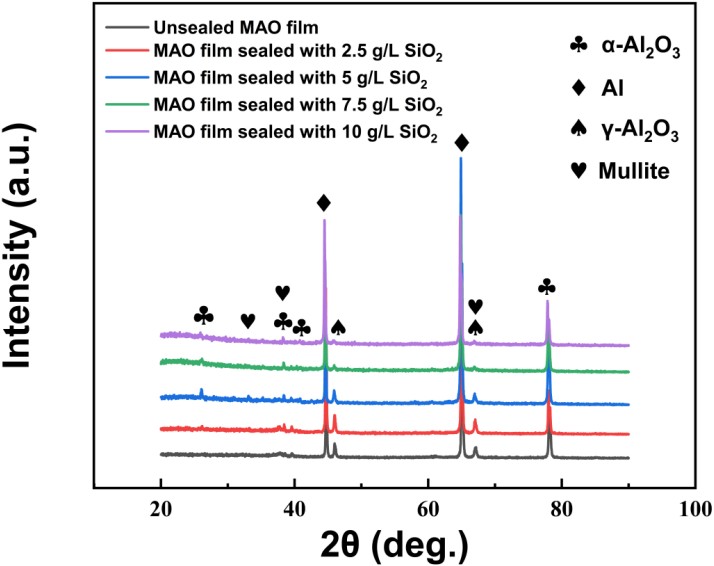

**Figure 4.** XRD patterns of MAO film and MAO films sealed by $SiO_2$ nanoparticles.

### 3.3. Mechanical Properties

Figure 5 shows the microhardness of unsealed MAO film and MAO film sealed by $SiO_2$ nanoparticles. The microhardness of unsealed MAO film was ~HV 1241. The microhardness of the MAO films increased significantly after sealing. When the concentration of $SiO_2$ nanoparticles in the electrolyte was 5 g/L, the microhardness of sealed MAO film reached a maximum of ~HV 1628, which was about 31.2% higher than that of the unsealed film. As the concentration of $SiO_2$ nanoparticles increased, the hardness of the sealed MAO film decreased. However, the hardness of the MAO films after sealing was always higher than that of the film without sealing.

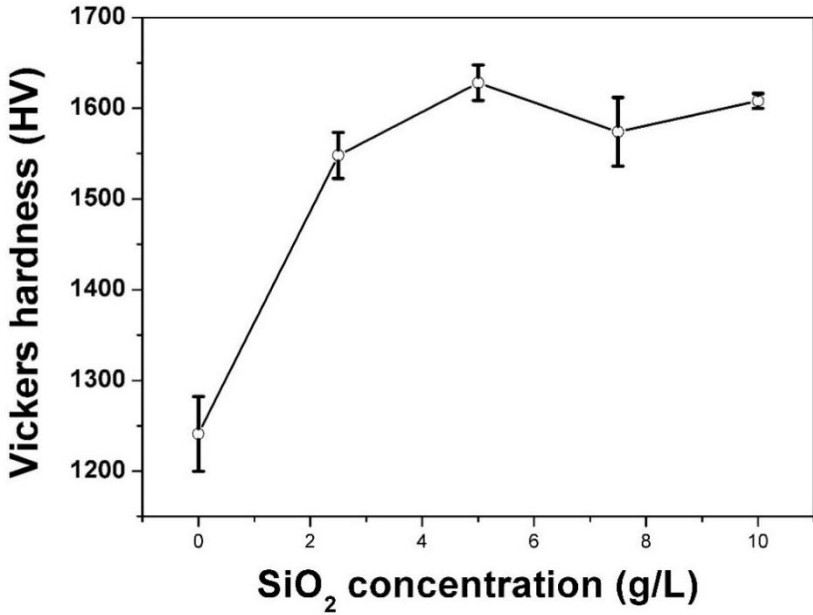

**Figure 5.** Microhardness of unsealed MAO film and MAO films sealed by $SiO_2$ nanoparticles.

The above results are related to the microstructure evolution of the MAO films. Appropriate addition of $SiO_2$ nanoparticles can effectively adsorb free ions and promote a more uniform voltage breakdown on the film surface. Consequently, the hardness of the surface ceramic layer was significantly improved due to the better crystallization of $Al_2O_3$ ejected from the homogeneously distributed holes.

### 3.4. Corrosion Resistance

Figure 6 shows the friction coefficient curves of different MAO films. The friction coefficient of the MAO film was more stable and fluctuated less than the matrix during the friction process. Table 4 presents the average friction coefficients of the substrate and the MAO films. The average friction coefficient of the MAO layer first decreased and then increased after sealing with $SiO_2$ nanoparticles. When the concentration of $SiO_2$ nanoparticles was 5 g/L, the average friction coefficient of the sealed film was 0.606, which was about 19.3% lower than the value of 0.773 for the matrix and 3.3% lower than the value of 0.627 for the unsealed film. This phenomenon was mainly attributed to the uniform surface structure and the reduced surface roughness after sealing. Further addition of $SiO_2$ nanoparticles gradually increased the film's average friction coefficient. Aggregation of nanoparticles aggravated the surface discharge of the film, resulting in the ablation of the film and the increase in surface roughness. When the concentration of $SiO_2$ nanoparticles was 10 g/L, the average friction coefficient of the film increased to 0.91.

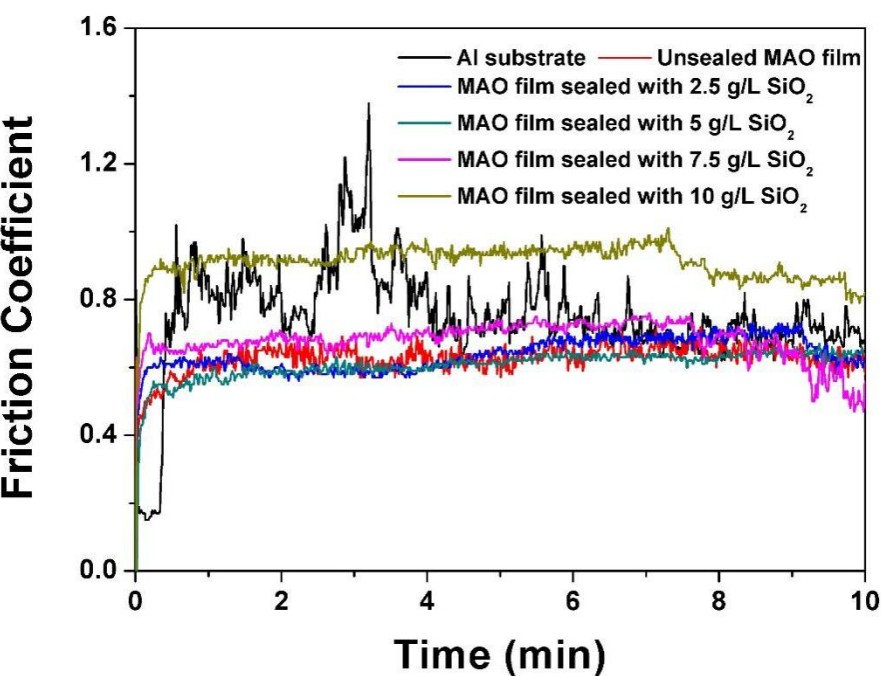

**Figure 6.** Wear track images of unsealed MAO film and MAO films sealed by SiO$_2$ nanoparticles.

**Table 4.** Average friction coefficient of different specimens.

| Sample | Average Friction Coefficient |
|---|---|
| Substrate | 0.773 |
| Unsealed MAO film | 0.627 |
| MAO film sealed with 2.5 g/L SiO$_2$ | 0.637 |
| MAO film sealed with 5 g/L SiO$_2$ | 0.606 |
| MAO film sealed with 7.5 g/L SiO$_2$ | 0.680 |
| MAO film sealed with 10 g/L SiO$_2$ | 0.910 |

Figure 7 shows a comparison of wear track widths for different samples. Table 4 presents the average friction coefficient of different specimens. The wear track width of unsealed MAO film was ~595 μm. After sealing with SiO$_2$ nanoparticles, the wear track width of the film was significantly reduced. Consistent with the variation trend of the average friction coefficient, the wear track width of the sealed film first decreased and then increased with increase in SiO$_2$ nanoparticle concentration. When the addition of SiO$_2$ nanoparticles was 5 g/L, the edge of the wear track was more uniform, and the wear track width decreased to ~425 μm. Due to the adsorption of nanoparticles on the film surface, further increase in the concentration of TiO$_2$ nanoparticles led to film ablation, increasing surface roughness and deterioration in wear resistance. When the concentration of SiO$_2$ nanoparticles was 10 g/L, the wear track width of the film rose to ~530 μm.

To investigate the effect of the sealing process on MAO film corrosion resistance, potentiodynamic polarization tests and electrochemical impedance spectroscopy measurements were performed on the samples. The polarization curves are shown in Figure 8. The fitting results of the polarization curves are shown in Table 5. E$_{corr}$, i$_{corr}$, β$_a$, β$_c$ were derived from the Tafel extrapolation method; Rp was calculated based on the Stern–Geary equation [41]. The corrosion potential of the films sealed with 5 g/L SiO$_2$ nanoparticles increased to −1.05 V compared to the unsealed films, which showed a significant passivation trend in the anodic region. This was attributed to the formation of a barrier layer after sealing with SiO$_2$ nanoparticles, which hindered the intrusion of the corrosive medium and thus prevented corrosion attack in the electrolyte. At the same time, the corrosion current density and corrosion rate decreased from $1.24 \times 10^{-9}$ A/cm$^2$ and $1.47 \times 10^{-5}$ mm/a to

$7.78 \times 10$–10 A/cm$^2$ and $9.15 \times 10^{-6}$ mm/a, respectively. The corrosion rate decreased by ~37%, indicating that the corrosion resistance of the sealed film was significantly improved.

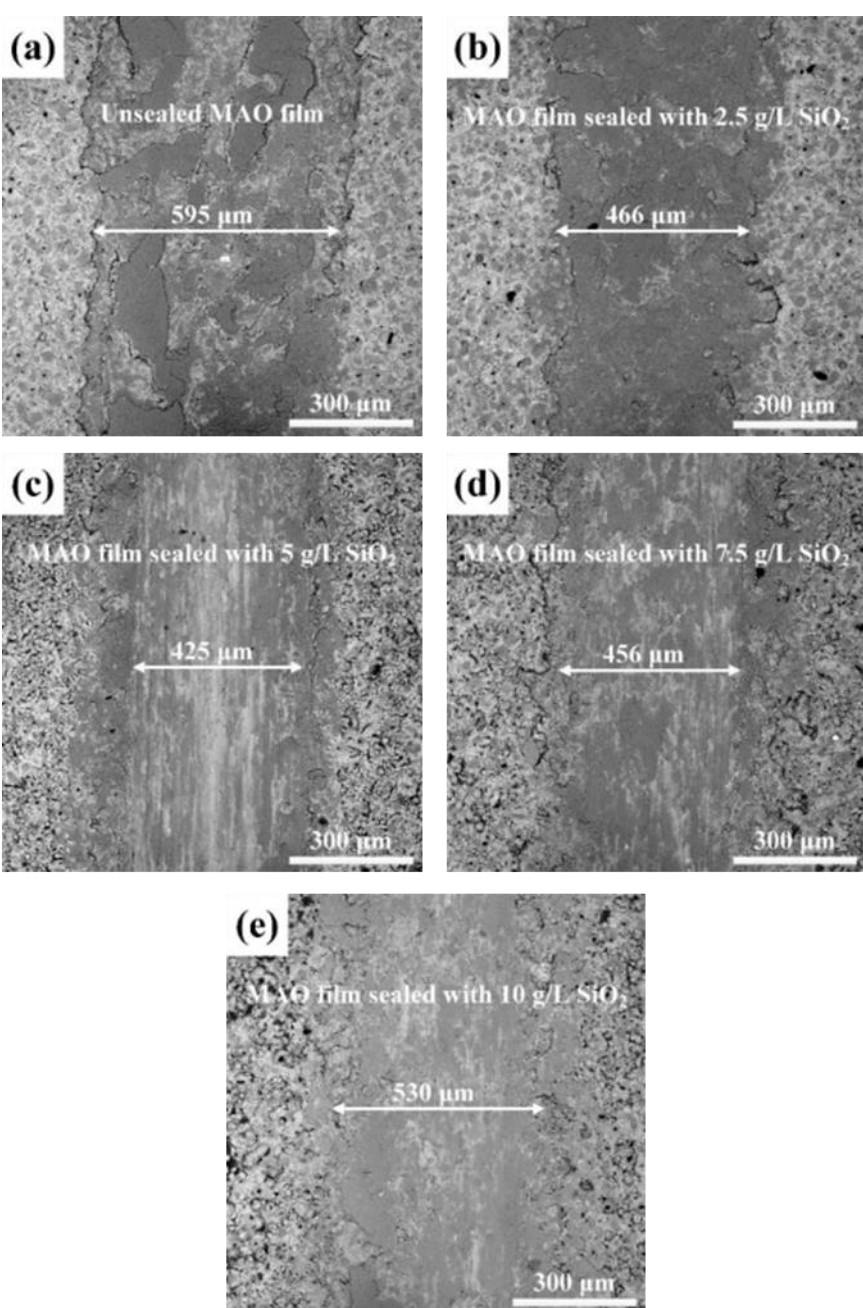

**Figure 7.** Wear track images of unsealed MAO film (**a**) and MAO films sealed by different concentrations of SiO$_2$ nanoparticles: (**b**) 2.5 g/L, (**c**) 5 g/L (**d**) 7.5 g/L and (**e**) 10 g/L.

Figure 9 shows the Nyquist plots of MAO film and MAO films sealed by SiO$_2$ nanoparticles. With increase in SiO$_2$ nanoparticle concentration in the electrolyte, the capacitive reactance arc radius of the film first increased and then decreased. When the concentration was 5 g/L, the impedance value reached a maximum, showing the best corrosion resistance. The increasing SiO$_2$ adsorbed the free ions in the solution under the discharge, resulting in excessive high surface energy, intensified discharge process, and finally, ablation. The molten oxide could not solidify uniformly on the film surface, which increased the cracks and reduced the corrosion resistance. Accordingly, the impedance value decreased to a minimum when the concentration of SiO$_2$ was 10 g/L.

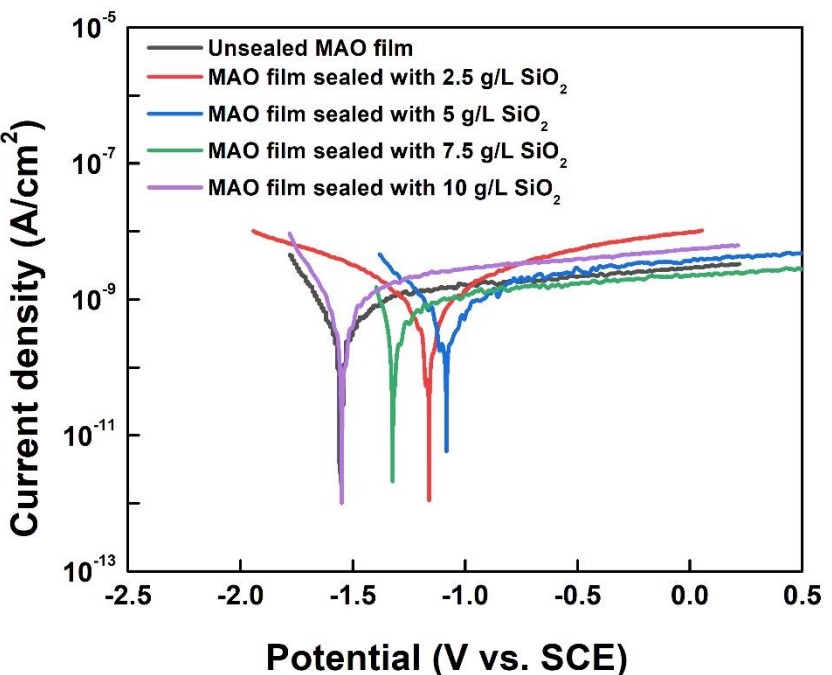

**Figure 8.** Polarization curves of unsealed MAO film and MAO films sealed by $SiO_2$ nanoparticles.

**Table 5.** Fitting results of polarization curves.

| Sample | $E_{corr}$ (V) | $I_{corr}$ (A/cm²) | Corrosion Rate (mm/a) | $B_a$ (mV/Decade) | $B_c$ (mV/Decade) | $R_p$ (Ω·cm²) |
|---|---|---|---|---|---|---|
| Unsealed MAO film | −1.56 | $1.24 \times 10^{-9}$ | $1.47 \times 10^{-5}$ | 559.09 | −238.59 | $5.88 \times 10^{6}$ |
| MAO film sealed with 2.5 g/L $SiO_2$ | −1.12 | $8.80 \times 10^{-10}$ | $9.37 \times 10^{-6}$ | 503.18 | −489.07 | $1.22 \times 10^{8}$ |
| MAO film sealed with 5 g/L $SiO_2$ | −1.05 | $7.78 \times 10^{-10}$ | $9.15 \times 10^{-6}$ | 638.93 | −368.26 | $1.31 \times 10^{8}$ |
| MAO film sealed with 7.5 g/L $SiO_2$ | −1.32 | $8.56 \times 10^{-10}$ | $9.87 \times 10^{-6}$ | 482.89 | −109.68 | $4.3 \times 10^{7}$ |
| MAO film sealed with 10 g/L $SiO_2$ | −1.55 | $1.15 \times 10^{-9}$ | $2.06 \times 10^{-5}$ | 504.01 | −223.34 | $5.8 \times 10^{7}$ |

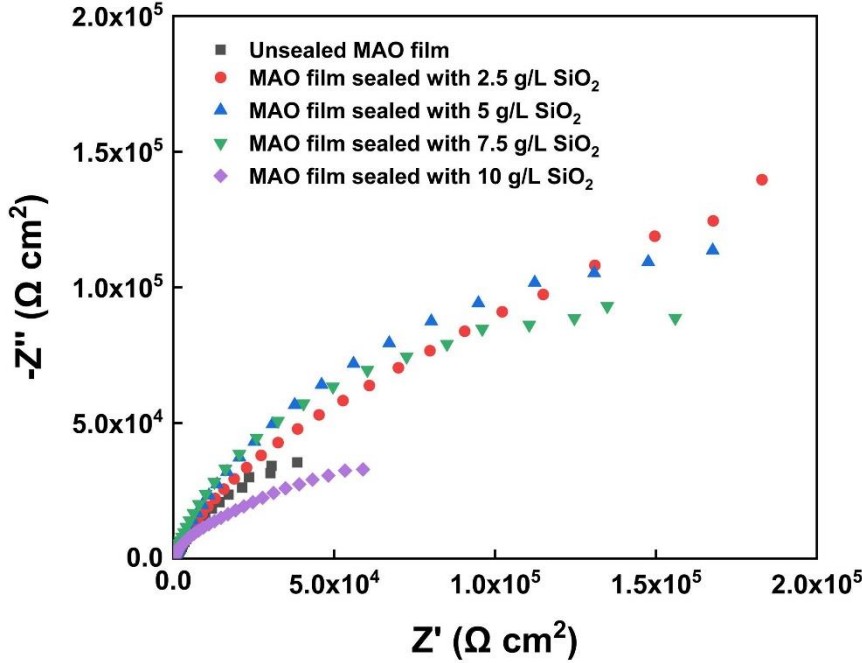

**Figure 9.** Nyquist plots of unsealed MAO film and MAO films sealed by $SiO_2$ nanoparticles.

Figure 10 shows the fitted equivalent circuit diagram of the electrochemical impedance spectrum, which consists mainly of two constant phase elements, since the MAO film usually consists of a dense inner layer and a sparse outer layer. $R_1$ is the solution resistance depending on the corrosive medium, and $CPE_1$ and $CPE_2$ are the constant phase elements, corresponding to the outer sparse layer and the dense inner layer, respectively. $R_{ct1}$ and $R_{ct2}$ represent the outer and inner film layer charge transfer resistance values (Table 6 shows the fitted parameters of each component). As can be seen from Table 6, the addition of $SiO_2$ nanoparticles increased the external Rct value by 1–2 orders of magnitude. When the concentration of $SiO_2$ nanoparticles was 5 g/L, the value of Rct1 reached its maximum value, which was about 11 times higher than that of the unsealed membrane layer. The Bode diagram reflects the impedance to a certain extent.

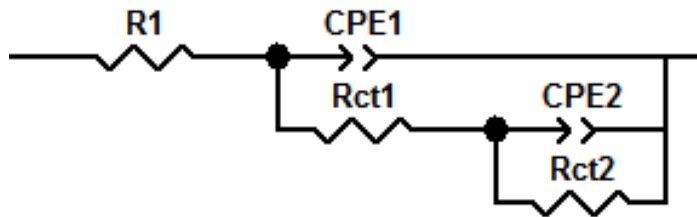

**Figure 10.** The equivalent circuit diagram of EIS.

**Table 6.** Equivalent circuit fitting parameters of EIS for different samples.

| Sample | $R_1$ ($\Omega \cdot cm^2$) | CPE1-T ($\Omega^{-1}\ cm^2\ s^{-P}$) | CPE1-P | $R_{ct1}$ ($\Omega \cdot cm^2$) | CPE2-T ($\Omega^{-1}\ cm^2\ s^{-P}$) | CPE2-P | $R_{ct2}$ ($\Omega \cdot cm^2$) |
|---|---|---|---|---|---|---|---|
| Unsealed MAO Film | 37.30 | $1.26 \times 10^{-6}$ | 0.62 | $9.93 \times 10^3$ | $4.75 \times 10^{-7}$ | 0.76 | $1.23 \times 10^5$ |
| MAO Film Sealed with 2.5 g/L $SiO_2$ | 65.40 | $1.50 \times 10^{-7}$ | 0.81 | $5.30 \times 10^4$ | $3.59 \times 10^{-7}$ | 0.71 | $4.91 \times 10^5$ |
| MAO Film Sealed with 5 g/L $SiO_2$ | 31.60 | $5.72 \times 10^{-8}$ | 0.88 | $1.13 \times 10^5$ | $5.80 \times 10^{-7}$ | 0.68 | $4.76 \times 10^5$ |
| MAO Film Sealed with 7.5 g/L $SiO_2$ | 50.80 | $1.49 \times 10^{-7}$ | 0.74 | $3.31 \times 10^4$ | $1.52 \times 10^{-7}$ | 0.89 | $3.31 \times 10^5$ |
| MAO Film Sealed with 10 g/L $SiO_2$ | 63.00 | $1.12 \times 10^{-7}$ | 0.89 | $1.12 \times 10^4$ | $5.53 \times 10^{-7}$ | 0.69 | $1.42 \times 10^5$ |

Figure 11 presents the Bode diagrams of unsealed MAO film and MAO films sealed by $SiO_2$ nanoparticles. The polarization resistance of the films after sealing was higher than that of the film without sealing as shown in Figure 11a. There were two time constants in the medium and high-frequency region of the unsealed film, as shown in Figure 11b. The unsealed film comprised a loose outer layer and a dense inner layer. When the unsealed film was in contact with the corrosive solution, the corrosive medium penetrated the porous outer layer and reacted with the compact inner layer. The time constant corresponding to the intermediate frequency region mainly derives from the contribution of the loose outer layer. In contrast, the time constant corresponding to the high frequency region primarily derives from the assistance of the dense inner layer. However, the phase angle diagram of the films after sealing has only one time constant in the high-frequency region. $SiO_2$ nanoparticles filled the micropores of the films, and the corrosive medium could not quickly penetrate into the inner layer, thus improving the corrosion resistance of the film.

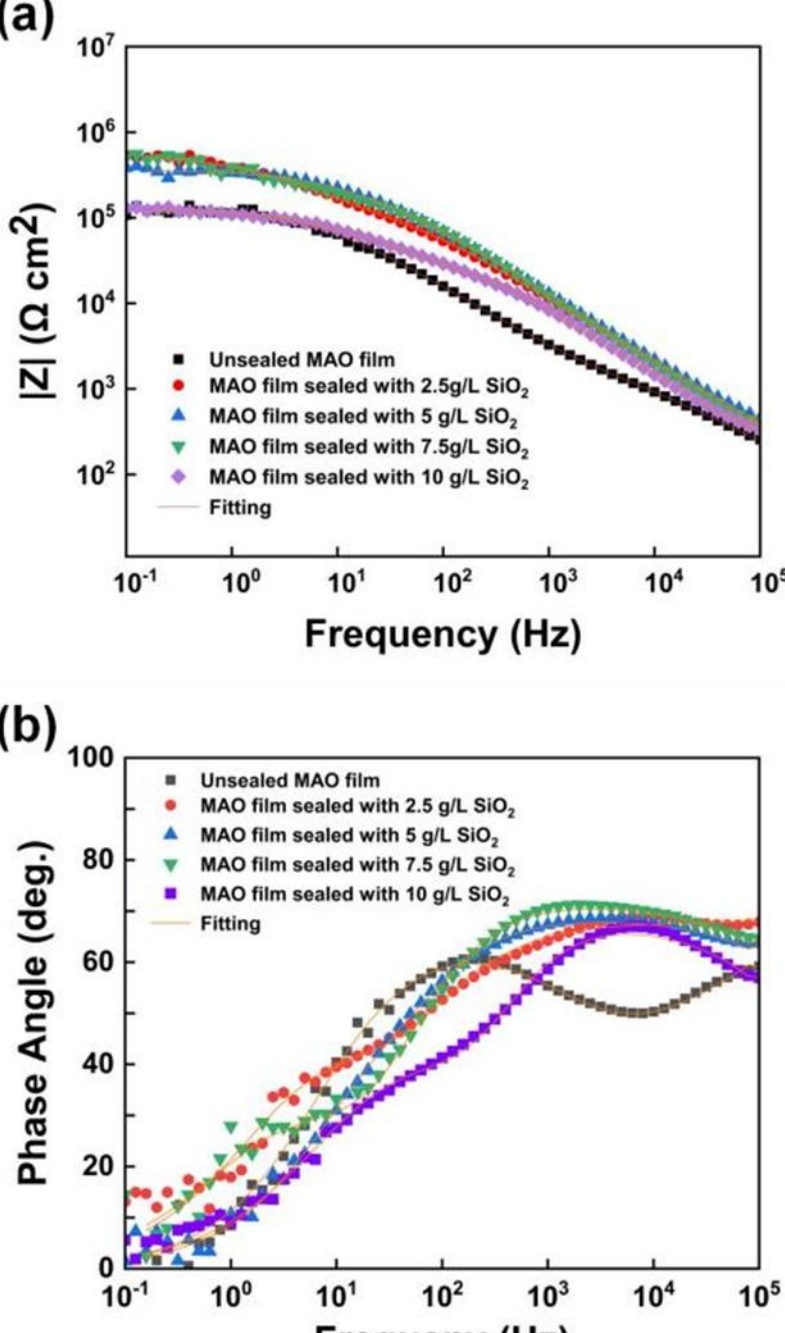

**Figure 11.** Bode plots of unsealed MAO film and MAO films sealed by $SiO_2$ nanoparticles: (**a**) relationship between frequency and Z; (**b**) relationship between frequency and phase angle.

## 4. Conclusions

The discharge micropores on the surface of MAO films can be uniformly filled and effectively sealed by $SiO_2$ nanoparticles. When the concentration of $SiO_2$ nanoparticles in the electrolyte was 5 g/L, the MAO films' microhardness, wear resistance, and corrosion resistance were significantly improved and reached an optimal value. Compared with unsealed film, the microhardness of the film surface increased by ~31.2%, while the wear track width decreased from ~595 μm to ~425 μm. In addition, the corrosion rate reduced by ~37%. The current research provides valuable information on the MAO processing of Al alloy and solid grounds for future industrial application.

**Author Contributions:** Conceptualization, S.L. and Y.W.; methodology, S.L. and Y.W.; formal analysis, S.L. and J.C.; investigation, J.C. and D.Z.; resources: Y.W. and P.G.; writing—original draft preparation: S.L.; writing—review and editing: Y.W. and Z.H.; supervision, Y.W. and Z.H.; project administration, Y.W. and Z.H. All authors have read and agreed to the published version of the manuscript.

**Funding:** The research was funded by the National Natural Science Foundation of China (51601073), the Research Project of Jiangsu Provincial Natural Science Fund (BK20211344), Jiangsu Provincial Six Talent Peaks (2018 XCL-028) and the Postgraduate Research and Practice Innovation Program of Jiangsu Province (KYCX21_3451).

**Institutional Review Board Statement:** Not applicable.

**Informed Consent Statement:** Not applicable.

**Data Availability Statement:** Not applicable.

**Acknowledgments:** The authors would like to express their appreciation to Chris Goode (Cirrus Materials Science Ltd., NZ) for his generous assistance.

**Conflicts of Interest:** The authors declare no conflict of interest.

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
