# Peer review of "Properties of Micro-Arc Oxidation Coatings on 5052 Al Alloy Sealed by SiO2 Nanoparticles"

_coatings, doi:10.3390/coatings12030373_

Round 1

Reviewer 1 Report

This manuscript deals with MAO of Al alloy in alkaline electrolyte with SiO2 particles as an additive. Anti-corrosion and mechanical performance have been evaluated. The work is of interest for readers of Coating journal, however, requires some major corrections listed below. The main concern is about absence of cross sectional characterisation of the coatings, neither thickness was provided as well. Without such basic characterisation, the paper has very low scientific value and may not be published.

  1. Wikipedia says that 5052 alloy includes Si, but experimental section does not mention any content of it.
  2. Electrical regime was not properly described. If it was unipolar, why have two duty cycles been mentioned? If it was bipolar, what were the current densities for positive and negative pulses?
  3. Thickness of the coating was not reported.
  4. Cross sectional SEM images are also necessary for publication. This is a very basic characteristics.
  5. Fig. 2 – should be replaced with table of values. For such standard composition, the spectra are not necessary to be presented.
  6. Fig. 3. XRD pattern is too small. Peak labels (symbols) are very difficult to distinguish.
  7. Fig. 4. It is unclear, how authors had measured the microhardness for such rough coatings? Was it polished? If yes, to which depth?

Author Response

Thanks for your insightful comments and valuable suggestions, which will help us to ameliorate our research paper. The suggested changes have been implemented in the revised manuscript, and major changes are highlighted based on the reviewer's comments.

  1. Wikipedia says that 5052 alloy includes Si, but experimental section does not mention any content of it.

Response: Thanks for your comment, and we feel really sorry for our negligence. The changes have been made to specify the 5052 aluminum alloy compositions (including Si) in the revised manuscript in Table 1.

  1. Electrical regime was not properly described. If it was unipolar, why have two duty cycles been mentioned? If it was bipolar, what were the current densities for positive and negative pulses?

Response: Thank you very much for your question. We used the bidirectional pulse current. The positive and negative current density was 6A/dm2. The positive and negative duty cycles were 30% & 40%.

  1. Thickness of the coating was not reported. Cross sectional SEM images are also necessary for publication. This is a very basic characteristics.

Response: Thank you for your comment. The cross-sectional characterization has been conducted. The results are presented in Fig. 2 and Fig. 3 as highlighted in the revised manuscript, supporting our statement that SiO2 helps seal the MAO film and enhance the film performance.

  1. Fig. 2 – should be replaced with table of values. For such standard composition, the spectra are not necessary to be presented.

Response: Thank you very much for your advice. We delete the old Fig. 2 and include the new Table 3 that describes the composition details of the prepared film, as the review suggested.

  1. Fig. 3. XRD pattern is too small. Peak labels (symbols) are very difficult to distinguish

Response: Thanks for your comment, and sorry for our negligence. The revised manuscript implemented the suggested changes to the XRD patterns in Fig. 4.

  1. Fig. 4. It is unclear, how authors had measured the microhardness for such rough coatings? Was it polished? If yes, to which depth?

Response: Thank you very much for your question. We illustrated the testing method in the experimental sections. A semi-automatic microhardness tester (HXS-1000TAC, Highwell, China) was used to test the surface hardness. The load was 10 N, with a loading time of 15 s and a holding time of 10 s. The resultant indentation was observed in SEM, based on which the microhardness value was calculated.

It is noted that no polishing treatment was performed, since we focused on the properties of the original film surface.

Reviewer 2 Report

In abstract, you need to provide quantitative results and avoid to mention the sentences like better resistance or the best resistance or similar non-scientific words.

In introduction section, you need to describe how you manage distribution of nanoparticles inside the coating and how you choose the concentration of nanoparticles. There are some reports which described the effects of nanoparticles on growth kinetics and their distribution inside the layer. Those papers must be reviewed and novelty of your work must be discussed at the end of introduction section. Some missed refs are: 10.1016/j.surfcoat.2016.08.055 , 10.1016/j.surfcoat.2016.07.074 , 10.1016/j.matchemphys.2016.10.019 , 10.1016/S1003-6326(18)64659-X , 10.1016/j.surfcoat.2016.07.086 , 10.1016/j.surfcoat.2019.125252 , 10.1016/j.surfcoat.2019.03.023 , 10.1007/s10800-008-9575-6

Usually sealing is not used for using nanoparticles in MAO coating. Sealing is useful post-treatment of layer with hot water or even application of post coatings such as organic coatings. In case of nanoparticles, usually we use composite coating.

The corrosion behavior of aluminium and also MAO coating on the surface of Al substrate: it is important to discuss the localized corrosion and the behavior of the anodic branch of polarization curves to reveal the effect of the coating and adding nanoparticles on localized corrosion resistance.

Author Response

Thanks for your insightful comments and valuable suggestions, which will help us to ameliorate our research paper. The suggested changes have been implemented in the revised manuscript, and major changes are highlighted based on the reviewer's comments.

  1. In abstract, you need to provide quantitative results and avoid to mention the sentences like better resistance or the best resistance or similar non-scientific words.

Response: The summary section has been revised with a more quantitative conclusion as suggested by the reviewer.

  1. In introduction section, you need to describe how you manage distribution of nanoparticles inside the coating and how you choose the concentration of nanoparticles. There are some reports which described the effects of nanoparticles on growth kinetics and their distribution inside the layer. Those papers must be reviewed and novelty of your work must be discussed at the end of introduction section. Some missed refs are: 10.1016/j.surfcoat.2016.08.055 , 10.1016/j.surfcoat.2016.07.074 , 10.1016/j.matchemphys.2016.10.019 , 10.1016/S1003-6326(18)64659-X , 10.1016/j.surfcoat.2016.07.086 , 10.1016/j.surfcoat.2019.125252 , 10.1016/j.surfcoat.2019.03.023 , 10.1007/s10800-008-9575-6

Response: Thanks for your suggestions. We have reviewed the proposed references and correctly cited them in the revised manuscript.

  1. Usually sealing is not used for using nanoparticles in MAO coating. Sealing is useful post-treatment of layer with hot water or even application of post coatings such as organic coatings. In case of nanoparticles, usually we use composite coating.

Answer: Thank you very much for your question. Despite the benefits, other treatment methods have some limits. For instance, the hydration sealing will consume the oxide film on the surface and decrease the film thickness. Organic sealing mainly uses physical adsorption, which has disadvantages such as poor bonding between organic and metal. In recent years, incorporating nanoparticles into MAO coatings by methods such as electrophoretic deposition (EPD) has become a popular research topic for improving their properties. By adding nanoparticles to an appropriate electrolyte system, the combined process of MAO and EPD could be accomplished. The MAO film with nanoparticle-sealed pores can be efficiently prepared at a low cost. (details in the reference: doi:10.1016/j.surfcoat.2016.07.086)

  1. The corrosion behavior of aluminium and also MAO coating on the surface of Al substrate: it is important to discuss the localized corrosion and the behavior of the anodic branch of polarization curves to reveal the effect of the coating and adding nanoparticles on localized corrosion resistance.

Response: Thank you for your advice. The plots show a significant passivation tendency in the anodic region. This is attributed to the formation of a barrier layer after the sealing with SiO2 nanoparticles. This hinders the penetration and prevents the corrosion attack.

Reviewer 3 Report

The manuscript submitted by Liu et al. can be accepted after major revision. The comments as follows;

  1. The MAO coatings were formed in complex electrolytes containing Na2SiO3, KOH, Na2WO4.2H2O without and with different concentrations of SiO2. The reason for selection such electrolyte should be explained.
  2. The absence of W-containing compounds in the XRD patterns should be explained. It is well known that tungsten oxide can be incorporated by decomposition of tungstate anions during MAO.
  3. The size of SiO2 nanoparticles should be indicated in the manuscript. The incorporation mechanism of SiO2 should be discussed as well.
  4. In contrast to the results reported in https://doi.org/10.1016/j.jallcom.2016.11.098, no peaks of SiO2 are identified?. Please add more explanations.
  5. MAO coating is usually composed of outer and inner layers. This means that at least two constant phase elements are required for fitting EIS results. However, the circuit model shown in Fig.1 is only composed of one-constant phase element. What is the reason?. The EIS results should be refitted using general circuit model to explore the protection properties provided by the outer and inner layers of MAO coatings. The present circuit model is only correct for fitting EIS results of the bare substrate. The cross-section images are suggested to be included.
  6. Bode plots should contain the fitting curves.
  7. The method used for calculation icorr should be clearly discussed. And Tafel slopes are needed to be included in the Table 4. Please not that the linear regions in some curves are absent.
  8. Some references are suggested to be included; for example the incorporation of other metal oxides into the MAO coatings made on Al alloy can be compared with the present results. For example:

(a) https://doi.org/10.1016/j.apsusc.2015.05.107

(b) https://iopscience.iop.org/article/10.1149/2.0931609jes/meta

(c) https://doi.org/10.1016/j.jmatprotec.2009.04.004

(d) https://doi.org/10.1016/j.ceramint.2015.12.136

Author Response

Thanks for your insightful comments and valuable suggestions, which will help us to ameliorate our research paper. The suggested changes have been implemented in the revised manuscript, and major changes are highlighted based on the reviewer's comments.

  1. The MAO coatings were formed in complex electrolytes containing Na2SiO3, KOH, Na2WO4.2H2O without and with different concentrations of SiO2. The reason for selection such electrolyte should be explained.

Response: Thanks for your comment. The silicate-based alkaline electrolyte is safe and environment-friendly. The Al3+ produced from the anode will migrate to the substrate surface, and then react with OH- in the electrolyte to form Al(OH)3 intermediates that are further converted to Al2O3 under the high voltage treatment. Therefore, silicate-based electrolyte is selected. The main components (Na2SiO3, KOH, Na2WO4.2H2O) and electrolyte concentrations were determined based on preliminary literature review and experiments.

  1. The absence of W-containing compounds in the XRD patterns should be explained. It is well known that tungsten oxide can be incorporated by decomposition of tungstate anions during MAO.

Response: The low concentration of Na2WO4.2H2O in the electrolyte (1.0 g/L) leads to limited W content in the deposits, which cannot be detected in XRD characterizations. Early studies have widely reported the inaccuracy of XRD testing for minor elements. In the meantime, the EDS analysis shows the existence of W in the MAO film. This is consistent with the conclusion that the decomposition of tungstate anion can incorporate tungsten oxide during the MAO process.

  1. The size of SiO2 nanoparticles should be indicated in the manuscript. The incorporation mechanism of SiO2 should be discussed as well.

Response: The coating performance improves when adding different concentrations (0-10 g/L) of nano-SiO2 particles (particle size ~20 nm) to the silicate electrolyte. The SiO2 nanoparticles can be uniformly deposited with well-dispersion by electrophoretic deposition (EPD) during the MAO process. The details could be found in the reference: (doi: 10.1016/j.surfcoat.2016.07.086). Our study sealed the MAO film on 5052 alloy, which greatly enhances its performance.

  1. In contrast to the results reported in

https://doi.org/10.1016/j.jallcom.2016.11.098, no peaks of SiO2 are identified?. Please add more explanations.

Response: Thank you very much for your question. Early studies have widely reported the inaccuracy of XRD testing for minor elements. Also, it is noted that different testing parameters were applied in the mentioned research. The SiO2 nanoparticles concentrate on the film surface with a limited incorporated content, confirmed by our more-accurate EDS results. The X-ray generally penetrates deeper in the coating with more information from the inner areas with fewer nanoparticles, thereby no SiO2 peak was observed in our study.

  1. MAO coating is usually composed of outer and inner layers. This means that at least two constant phase elements are required for fitting EIS results. However, the circuit model shown in Fig.1 is only composed of one-constant phase element. What is the reason?. The EIS results should be refitted using general circuit model to explore the protection properties provided by the outer and inner layers of MAO coatings. The present circuit model is only correct for fitting EIS results of the bare substrate. The cross-section images are suggested to be included.

Response: Thank you very much for your advice. The equivalent circuit model is revised in Fig. 10 of the revised manuscript. The related description has been added as highlighted in Section 3.3. Moreover, the cross-sectional surface morphology and EDS results have been included in Fig. 2 and Fig. 3, which illustrates the inner and outer part of the prepared MAO films.

  1. Bode plots should contain the fitting curves.

Response: Thank you very much for your advice. The corresponding figures have been modified, and the fitted curve has been included.

  1. The method used for calculation icorr should be clearly discussed. And Tafel slopes are needed to be included in the Table 4. Please note that the linear regions in some curves are absent.

Response: Ecorr, icorr, βa, βc are derived from the Tafel extrapolation method Rp is calculated based on the Stern-Geary equation (see reference: DOI: 10.1149/1.2428473). The revalent paramters have been revised as shown in Table 5.

  1. Some references are suggested to be included; for example the incorporation of other metal oxides into the MAO coatings made on Al alloy can be compared with the present results. For example:

(a) https://doi.org/10.1016/j.apsusc.2015.05.107

(b) https://iopscience.iop.org/article/10.1149/2.0931609jes/meta

(c) https://doi.org/10.1016/j.jmatprotec.2009.04.004

(d) https://doi.org/10.1016/j.ceramint.2015.12.136

Response: Thanks for your suggestions. We have reviewed the proposed references carefully and correctly cited them in the revised manuscript.

Round 2

Reviewer 1 Report

We used the bidirectional pulse current. The positive and negative current density was 6A/dm2. The positive and negative duty cycles were 30% & 40%. – this must be in the text, but was not found.

Reviewer 2 Report

accept

Reviewer 3 Report

It can be accepted in the present form. However, the cathodic Tafel slopes (Table 5) should be negative, So, please revise them during proof-reading